# Impact of Individual Comorbidities on Survival of Patients with Myelofibrosis

**DOI:** 10.3390/cancers14092331

**Published:** 2022-05-09

**Authors:** María García-Fortes, Juan C. Hernández-Boluda, Alberto Álvarez-Larrán, José M. Raya, Anna Angona, Natalia Estrada, Laura Fox, Beatriz Cuevas, María C. García-Hernández, María Teresa Gómez-Casares, Francisca Ferrer-Marín, Silvana Saavedra, Francisco Cervantes, Regina García-Delgado

**Affiliations:** 1Hematology Department, Hospital Universitario Virgen de la Victoria, 29010 Málaga, Spain; regina.garcia.delgado.sspa@juntadeandalucia.es; 2Facultad de Medicina, Universidad de Málaga, 29010 Málaga, Spain; 3Hematology Department, Hospital Clínico, INCLIVA, 46010 Valencia, Spain; hernandez_jca@gva.es; 4Hematology Department, Hospital Clínic, 08036 Barcelona, Spain; aalvar@clinic.cat (A.Á.-L.); fcervan@clinic.ub.es (F.C.); 5Hematology Department, Hospital Universitario de Canarias, 38320 Santa Cruz de Tenerife, Spain; jrayasan@ull.es; 6Hematology Department, Hospital del Mar, 08003 Barcelona, Spain; aangona@parcdesalutmar.cat; 7Hematology Department, ICO-Hospital Germans Trias i Pujol, Josep Carreras Leukemia Research Institute, 08916 Badalona, Spain; nestrada@carrerasresearch.org; 8Hematology Department, Vall d’Hebron Institute of Oncology (VHIO), Vall d’Hebron Hospital Universitari, 08035 Barcelona, Spain; mlfox@vhebron.net; 9Hematology Department, Hospital Universitario de Burgos, 09006 Burgos, Spain; bcuevas@hubu.es; 10Hematology Department, Hospital General Universitario de Alicante, 03010 Alicante, Spain; garcia_mcar@gva.es; 11Hematology Department, Hospital Universitario Doctor Negrín, 35010 Las Palmas de Gran Canaria, Spain; mgomcasf@gobiernodecanarias.org; 12Hematology Department, Hospital General Universitario Morales Meseguer, CIBERER, IMIB, UCAM, 30008 Murcia, Spain; fferrer@ucam.edu; 13Hematology Department, Hospital Santa Creu i Sant Pau, 08025 Barcelona, Spain; ssaavedrag@santpau.cat

**Keywords:** myelofibrosis, comorbidities, survival, prognosis

## Abstract

**Simple Summary:**

The coexistence of cancer with other chronic conditions has substantial implications for treatment decisions and outcomes for both neoplasms and chronic disease. Reports have demonstrated the impact of comorbidities on survival in different hematologic disorders. Myelofibrosis (MF) guidelines do not consider the complex interrelations between MF and comorbidity. Several works have shown how MF patients have a wide variety and high burden of comorbidities and demonstrated that the comorbidity burden was significantly associated with an unfavorable impact on survival. These previous studies about comorbidity on MF are retrospective and consider the cumulative rather than individual comorbidity burden. The influence of individual comorbidities on outcome in MF patients has not been studied. We sought to identify the comorbidities in MF patients at diagnosis and to assess the influence of those different comorbidities on survival. Considering them individually may contribute to the personalization of MF management and optimizing outcomes.

**Abstract:**

The comorbidity burden is an important risk factor for overall survival (OS) in several hematological malignancies. This observational prospective study was conducted to evaluate the impact of individual comorbidities on survival in a multicenter series of 668 patients with primary myelofibrosis (PMF) or MF secondary to polycythemia vera (PPV-MF) or essential thrombocythemia (PET-MF). Hypertension (hazard ratio (HR) = 4.96, *p* < 0.001), smoking (HR = 5.08, *p* < 0.001), dyslipidemia (HR = 4.65, *p* < 0.001) and hepatitis C virus (HCV) (HR = 4.26, *p* = 0.015) were most adversely associated with OS. Diabetes (HR = 3.01, *p* < 0.001), pulmonary disease (HR = 3.13, *p* < 0.001) and renal dysfunction (HR = 1.82, *p* = 0.037) were also associated with an increased risk of death. Multivariate analysis showed that pulmonary disease (HR = 2.69, *p* = 0.001), smoking (HR = 3.34, *p* < 0.001), renal dysfunction (HR = 2.08, *p* = 0.043) and HCV (HR = 11.49, *p* = 0.001) had a negative impact on OS. When ruxolitinib exposure was included in the model, the effect of each comorbidity on survival was modified. Therefore, individual comorbidities should be taken into account in determining the survival prognosis for patients with MF.

## 1. Introduction

Myelofibrosis (MF), either primary (PMF) or evolving after essential thrombocythemia MF (PET-MF) or polycythemia MF (PPV-MF), is a Philadelphia chromosome-negative myeloproliferative neoplasm (MPN) characterized by the mobilization of clonal hematopoietic cells from the fibrotic bone marrow to extramedullary organs, mainly the spleen and liver [1,2].

The estimated incidence of PMF ranges from 0.1 to 1.5 cases per 100,000 individuals per year, with a peak incidence in the sixth decade of life. The risk of progression to MF in polycythemia vera or essential thrombocythemia at 15 years is 6–14% and 4–11%, respectively [3,4,5,6].

About 90% of patients with MF present mutations in the JAK2, CALR or MPL genes, which activate the JAK/STAT signaling pathway, resulting in increased levels of inflammatory cytokines that may trigger constitutional symptoms [7,8,9]. Disease manifestations include progressive anemia, splenomegaly caused by extramedullary hematopoiesis and constitutional symptoms such as fatigue, itching, night sweats, low grade fever, weight loss and bone pain [10,11].

In clinical practice, treatment options include supportive care, cytoreductive agents or JAK inhibitors [12]. The only curative option for MF is allogeneic hematopoietic stem cell transplant (HSCT), but this procedure is restricted to a minority of patients with MF due to its significant morbidity and mortality [13,14]. Various prognostic scoring systems have been developed to predict the survival of MF patients [15,16,17,18,19,20,21,22]. However, comorbidities have not been included as risk factors in any of these prognostic models [23,24] despite their influence on survival in other hematological malignancies such as chronic myeloid leukemia [25], chronic lymphocytic leukemia [26], myelodysplastic syndromes [27], and acute myeloid leukemia [28] or HSCT [29].

Previous studies have shown that the coexistence of cancer and other chronic conditions has substantial implications for treatment decisions and outcomes for both neoplasms and chronic disease [30]. MF is a disabling condition mainly affecting elderly people [4,5,6]. Comorbidities are more prevalent in elderly patients and can potentially interact with the MF phenotype to increase long-term mortality [31,32,33,34].

Although few studies have assessed the impact of comorbidities on the outcome of patients with MF, retrospective studies have shown that a high burden of comorbidities as defined by the Adult Comorbidity Evaluation-27 (ACE-27) [31] is significantly associated with an unfavorable impact on survival in MF [32,33,34]. However, these studies considered the cumulative comorbidity burden rather than the impact of individual comorbidities on survival.

The main aim of the present study was to assess the influence of individual comorbidities on survival in a prospective multicenter series of MF patients.

## 2. Materials and Methods

### 2.1. Study Design

This observational prospective study was based on data obtained from a national multicenter MF register sponsored by the Spanish MPN Group (GEMFIN). The study design consisted of a baseline visit between February 2014 and October 2018 followed by two follow-ups at 6 and 12 months after inclusion. Each patient’s survival was monitored until death. Interviews, complementary tests and the treatment plan were carried out in line with routine clinical practice.

The inclusion criteria applied were that the patients should be aged ≥18 years, be diagnosed with MF (PMF, PPV-MF or PET-MF) as defined by the World Health Organization [1,2], and voluntarily provide signed informed consent.

The study data were recorded on a purpose-designed electronic case report form, which included standardized demographic, disease and treatment information and provided confidentiality, security and authenticity. The presence of individual comorbidities was considered at diagnosis (see the Table A1 for a list of the definitions of comorbidities). Risk stratification at diagnosis was performed using the International Prognostic Scoring System (IPSS) [15]. MF symptoms at the baseline visit were assessed by MPN-SAF [35]. This study was based on a prior review protocol (GEM-MIE-2014-01) approved by local ethics committees, and all research was performed in accordance with the provisions of the Declaration of Helsinki.

### 2.2. Statistical Analysis

The main endpoint was the impact of individual comorbidities on overall survival (OS) in patients with MF. Comorbidities were considered if they were present at the time of MF diagnosis. OS was calculated from the date of diagnosis until the date of death or the last follow-up (censored).

Univariate descriptive statistics of the study population were calculated. Quantitative variables were described by means of centralization and dispersion measures, and categorical data were presented as absolute numbers (N) with percentages (%).

Parametric (*t*-test) or non-parametric (Mann–Whitney) statistical tests were performed as appropriate to compare two independent means. According to the sample distribution, parametric (paired *t*-test) or non-parametric (Wilcoxon) statistical tests were performed to compare paired means. Differences in the distributions of categorical variables were evaluated by the chi-square test.

The evolution of a qualitative variable between two time points (before–after) was determined, and the associated *p*-value obtained, by the McNemar test.

Survival probability was determined by the Kaplan–Meier method. Multivariate adjusted hazard ratios for prognostic factors were estimated by Cox’s proportional hazard regression model. A *p*-value < 0.05 was deemed to indicate statistical significance. Variables with a *p*-value ≤ 0.05 in the univariate analysis (age (categorized into three groups defined by quartiles), hypertension, diabetes, dyslipidemia, pulmonary disease, renal dysfunction, smoking, IPSS, CALR gene mutation status and ruxolitinib treatment) were included in the multivariable analysis. Since ruxolitinib is not an intrinsic characteristic of the patient, but an external treatment that the patient may receive according to medical criteria, the model was applied with and without ruxolitinib.

All statistical analyses were performed using SPSS v24.0 (Dynamic, Madrid, Spain).

## 3. Results

### 3.1. Patient Characteristics

The study population was composed of 668 patients. Table 1 shows their demographic characteristics, disease history and comorbidities at diagnosis of MF. The median age of the patients at diagnosis was 68 years (range 25–89), 244 (36.5%) were female, and 61% had PMF. JAK2 and CALR mutations were present in 56.1% and 9.1% of these patients, respectively. On the IPSS 15, 64% (*n* = 431) were classed as intermediate-2 or high risk.

The most common comorbidities were hypertension (*n* = 282, 42.2%), smoking (*n* = 161, 24.1%), diabetes (*n* = 124, 18.6%), dyslipidemia (*n* = 117, 16%) and cardiovascular disease (*n* = 105, 15.7%). Other comorbidities, which were less frequent but were observed in ≥5% of the patients, were pulmonary disease (*n* = 55, 8.2%), renal dysfunction (*n* = 58, 8.7%) and other neoplasms (*n* = 55, 8.2%). The prevalence of each comorbidity is detailed in Table 1.

The median time elapsed from MF diagnosis to inclusion in the study was 5.9 years. At the time of inclusion, 15.7% (*n* = 105) of the patients were receiving ruxolitinib treatment, and the median MPN-SAF37 score was 19 (range 0–62).

### 3.2. Impact of Risk Factors on Survival

After a median follow-up of 2.49 years, 380 (56.8%) of the patients had died. The median survival was 4.01 years (3.45–4.57). Table 2 shows the results of the univariate analysis performed. Survival was greater among patients younger than 61.4 years than among those aged 61.4–76.3 years (HR = 2.04; 95% confidence interval (CI), 1.54–2.69, *p* < 0.001) and 76.3 years (HR = 4.43; 95% CI, 3.23–6.06, *p* < 0.001). In our cohort, the female patients survived longer than males (HR = 0.76; 95% CI, 0.61–0.95, *p* = 0.017). Intermediate-2 (HR = 3.65; 95% CI, 2.31–5.77, *p* < 0.001) and high-risk IPSS categories were associated with shorter survival (HR = 5.10; 95% CI, 3.20–8.11, *p* < 0.001). By contrast, patients presenting a CALR mutation (HR = 0.49; 95% CI, 0.29–0.83, *p* = 0.009) or treated with ruxolitinib (HR = 0.04; 95% CI, 0.01–0.12, *p* < 0.001) had better odds of survival.

Hypertension (HR = 4.96; 95% CI, 3.26–7.55, *p* < 0.001), smoking (HR 5.08; 95% CI, 3.35–7.71, *p* < 0.001), dyslipidemia (HR 4.65; 95% CI, 3.11–6.95, *p* < 0.001) and HCV (HR 4.26; 95% CI, 1.32–13.75, *p* < 0.015) were all strongly associated with worse survival (HR > 4 for each factor). Diabetes (HR 3.01; 95% CI, 2.07–4.36, *p* < 0.001), pulmonary disease (HR 3.13; 95% CI, 1.86–5.26, *p* < 0.001) and renal dysfunction (HR 1.82; 95% CI, 1.04–3.19, *p* = 0.037) were significantly associated with an increased risk of death (Figure 1). Cardiovascular comorbidity and other neoplasms showed a trend toward worse survival, but the difference was not statistically significant (*p* = 0.186 and *p* = 0.052, respectively).

### 3.3. Multivariate Analysis

Table 3 shows the multivariate Cox’s proportional hazards models obtained. When ruxolitinib was excluded from the regression analysis, the individual comorbidities significantly associated with survival were pulmonary disease (HR = 2.69; 95% CI, 1.47–4.91, *p* = 0.001), smoking (HR = 3.34; 95% CI, 1.85–6.04, *p* < 0.001), renal dysfunction (HR = 2.08; 95% CI, 1.02–4.21, *p* = 0.043) and HCV (HR = 11.49; 95% CI, 2.74–48.25, *p* = 0.001).

Additional independent risk factors for survival were age and IPSS. Thus, shorter survival was associated with increasing age (age range 61.4–76.3 (HR = 2.51; 95% CI, 1.12–5.61, *p* = 0.026), age ≥76.3 (HR = 4.85; 95% CI, 1.75–13.41, *p* = 0.002)) and higher-risk IPSS categories (IPSS Intermediate-2 vs. low risk (HR = 4.76; 95% CI, 1.39–16.22, *p* = 0.013), IPSS high risk vs. low risk (HR = 11.34; 95% CI, 3.24–39.70, *p* < 0.001)).

When ruxolitinib was included in the regression analysis, the individual comorbidities significantly associated with survival were pulmonary disease (HR = 2.40; 95% CI, 1.29–4.47, *p* = 0.006), smoking (HR = 3.82; 95% CI, 2.02–7.24, *p* < 0.001) and HCV (HR = 9.86; 95% CI, 2.34–41.64, *p* = 0.002). Age 61.4 years and IPSS Intermediate-2 continued to have a negative impact on survival: age 61.4–76.3 (HR = 2.91; 95% CI, 1.24–6.85, *p* = 0.014), age ≥ 76.3 (HR = 4.23; 95% CI, 1.53–11.73, *p* = 0.006), IPSS Intermediate-2 vs. low risk (HR = 6.90; 95% CI, 1.95–24.37, *p* = 0.003) and IPSS high risk vs. low risk (HR = 15.97; 95% CI, 4.22–60.42, *p* < 0.001). Ruxolitinib treatment was significantly associated with better survival (HR = 0.13; 95% CI, 0.38–0.42, *p* = 0.001).

## 4. Discussion

As expected, the majority of patients in our cohort had comorbidities that ultimately affected their fitness and ability to undergo MF therapy. Current methods of risk-stratifying patients with MF, including widely used prognostic models such as IPSS [15], DIPSS [16] and DIPSS Plus [17], do not take into account patient comorbidities, although they are known to impact survival in numerous malignancies. Until recently, clinical trials excluded patients with significant organ dysfunction and thus provided limited information on how such patients should be managed. The aim of our prospective study was to investigate the influence of individual comorbidities at diagnosis on the outcomes of MF patients.

Comorbidities are known to be associated with inferior survival among patients with MF [32,33,34]. The ACE-27 [31], which is specifically designed for patients with cancer, is the instrument most commonly used in previous studies of MF patients to measure the severity of their comorbidities [32,33,34]. In 2014, a retrospective study of 131 MF patients by Lekovic et al. [34] suggested ACE-27 could help predict patient survival. On the other hand, their multivariate model did not find age to be an important prognostic factor for survival. In the same year, Newberry et al. [33] reported that comorbidities had a significant negative impact on OS in PMF patients aged <65 years, but not in older patients. More recently, Bartozsko et al. [32] evaluated two comorbidity scales in a cohort of 309 patients with PPV-MF or PET-MF and PMF, and concluded that severe comorbidities according to ACE-27 (score ≥ 3) were associated with a reduced OS. By contrast, a high score on the hematopoietic cell transplantation comorbidity index (HCT-CI) [36] did not reach statistical significance. No differential effect of severe comorbidities on survival based on age was detected. Finally, Breccia et al. [37] showed that baseline comorbidities did not influence the probability of achieving spleen/symptom responses for MF patients receiving ruxolitinib treatment. It should be noted that the assessment of comorbidities differed among these studies, which precluded our reaching firm conclusions on the impact of this factor on survival in MF.

In our study, smoking, pulmonary disease, HCV and renal comorbidities were independent predictors of OS, whereas diabetes was only marginally associated with worse survival in MF patients. By contrast, hypertension and dyslipidemia did not have a significant impact on outcomes. As expected, older age and advanced IPSS were associated with poor survival. Of note, ruxolitinib treatment was associated with improved survival and with the impact of comorbidities on OS. Thus, when ruxolitinib exposure was included in the regression model, renal dysfunction was no longer associated with survival, while other comorbidities such as HCV and pulmonary conditions had a weaker influence on OS. These findings might be due to the well-known anti-inflammatory effect of ruxolitinib [38], counteracting the inflammation process mediated by these comorbidities, as has been suggested by Hasselbalch [39,40].

Neither hypertension nor dyslipidemia significantly influenced outcomes, perhaps due to the effective management of these comorbidities (assuming the levels recorded were within the target range). Several recent studies on the impact of cardiovascular risk factors (CVRFs) such as diabetes, hypertension or dyslipidemia on cardiovascular complications have demonstrated significant improvement in outcomes achieved by early, effective control [41,42,43].

The risk of cardiovascular events is increased in MF patients [44,45], but it is unclear to what extent this is a direct complication of MF and what role is played by other CVRFs. The physiopathology of cardiovascular events is known to be associated with inflammatory disorders [46], but further research is needed to measure the impact of correct management of CVRFs and the effect of ruxolitinib on these events. Both CVRF control and ruxolitinib are believed to reduce levels of vascular inflammation [39,40] and could play a crucial role in alleviating or preventing inflammation-mediated complications. Therefore, close collaboration between the hematologist and other medical specialists in managing and controlling comorbid conditions is an important aspect of achieving optimal outcomes.

It has long been understood that comorbidities are a significant element in the evolution of cancer patients and that any evaluation should take this circumstance into account [30]. The cumulative burden of morbidity, rather than individual, disease-specific effects, is normally considered a risk marker for mortality [26,27,28,29,30,31,32,33,34]. To our knowledge, the present study is the first to consider the impact of individual comorbidities on patients with MF.

Our analysis has several limitations that should be acknowledged. First, the restricted sample size and event number per comorbidity limited the power to detect comorbidity effects. However, the major comorbidities were well-represented in the cohort. Second, our study considered all-cause and non-MF-related mortality versus other specific disease mortality. This approach was taken because it is difficult to ascertain disease-related deaths in an elderly population such as that of MF patients. Third, because the study was designed before the 2016 WHO classification, we did not make distinction between prefibrotic MF and overt MF [1]. Finally, the impact of comorbidities may change due to variations in the medical care provided.

## 5. Conclusions

In summary, our study findings suggest that individual comorbidities may significantly influence the OS of MF patients. Therefore, it is essential to take individual comorbidities into account in forecasting survival and optimizing treatment management for MF patients. Treatment with ruxolitinib seems to reduce the deleterious effect of specific comorbidities, which could partially explain its association with improved survival in MF patients. However, further studies are needed to confirm these findings.

## Figures and Tables

**Figure 1 cancers-14-02331-f001:**
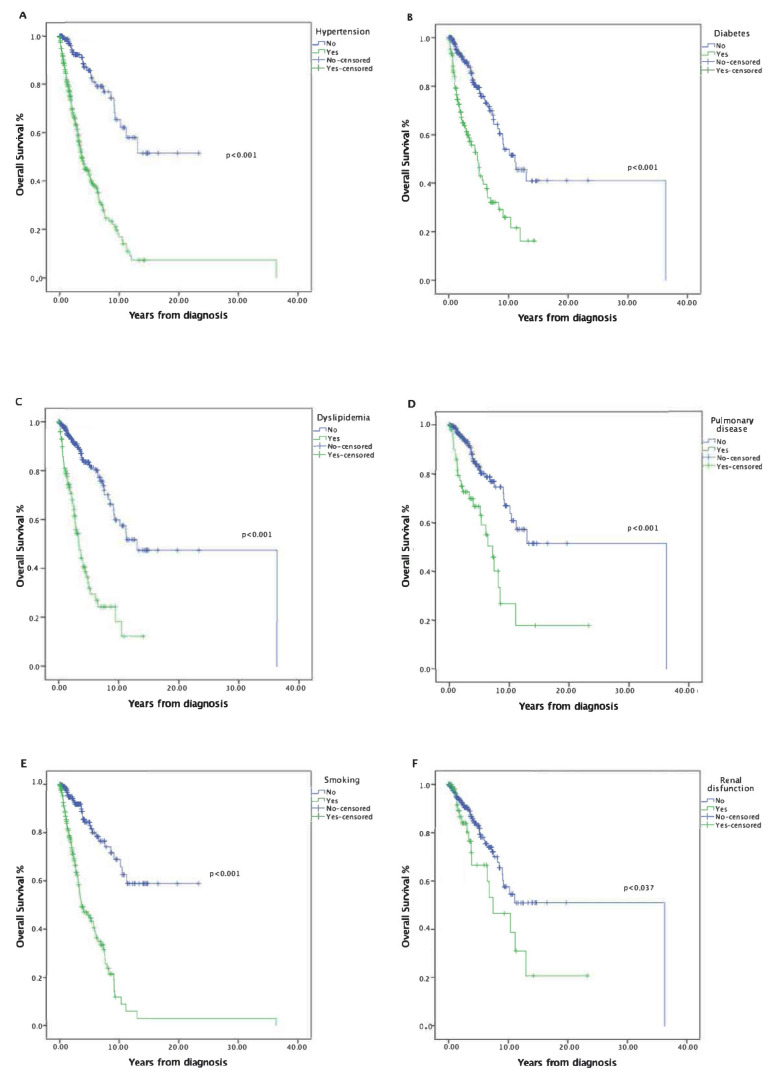
Kaplan-Meier analyses showing overall survival in patients with MF. (**A**) Univariate analysis for overall survival by hypertension. (**B**) Univariate analysis for overall survival by diabetes. (**C**) Univariate analysis for overall survival by dyslipidemia. (**D**) Univariate analysis for overall survival by pulmonary disease. (**E**) Univariate analysis for overall survival by smoking. (**F**) Univariate analysis for overall survival by renal dysfunction.

**Table 1 cancers-14-02331-t001:** Demographic data, patient comorbidities and disease characteristics at diagnosis of MF.

Variable	All Patients *n* = 668
Median age, years	68 (25–89)
Female, *n* (%)	244 (36.5)
Hypertension, *n* (%)	282 (42.2)
Diabetes, *n* (%)	124 (18.6)
Dyslipidemia, *n* (%)	117 (16.0)
Cardiovascular disease, *n* (%)	105 (15.7)
Pulmonary disease, *n* (%)	55 (8.2)
Renal dysfunction, *n* (%)	58 (8.7)
Hepatic disease, *n* (%)	41 (6.2)
HIV, *n* (%)	0 (0)
HBV, *n* (%)	20 (3)
HCV, *n* (%)	10 (1.5)
Other neoplasm, *n* (%)	55 (8.2)
Smoking, *n* (%)	161 (24.1)
PMF, *n* (%)	411 (61.5)
JAK2, *n* (%)	375 (56.1)
CALR, *n* (%)	61 (9.1)
Splenomegaly ^+^, *n* (%)	371 (55.5)
IPSS	
Low risk, *n* (%)	71 (10.6)
Intermediate 1, *n* (%)	166 (24.9)
Intermediate 2, *n* (%)	243 (36.4)
High risk, *n* (%)	188 (28.1)

HIV, human immunodeficiency virus. HBV, hepatitis B virus. HCV, hepatitis C virus. IPSS, International Prognostic Scoring System. PMF, primary myelofibrosis. ^+^ Splenomegaly determined by imaging methods such as ultrasonography or computed tomography.

**Table 2 cancers-14-02331-t002:** Univariate survival model.

Variable	HR	95% CI	*p*-Value
Age (61.4–76.3 y) *	2.04	(1.54–2.69)	<0.001
Age (≥76.3 y) *	4.43	(3.23–6.06)	<0.001
Female	0.76	(0.61–0.95)	0.017
Hypertension	4.96	(3.26–7.55)	<0.001
Diabetes	3.01	(2.07–4.36)	<0.001
Dyslipidemia	4.65	(3.11–6.95)	<0.001
Cardiovascular disease	1.41	(0.85–2.35)	0.186
Pulmonary disease	3.13	(1.86–5.26)	<0.001
Renal dysfunction	1.82	(1.04–3.19)	0.037
HCV	4.26	(1.32–13.75)	0.015
Other neoplasm	1.76	(0.99–3.12)	0.052
Smoking	5.08	(3.35–7.71)	<0.001
PMF vs. PPV-MF or ET-MF	1.07	(0.86–1.33)	0.534
IPSS Intermediate-1 **	1.56	(0.95–2.55)	0.740
IPSS Intermediate-2 **	3.65	(2.31–5.77)	<0.001
IPSS High Risk **	5.10	(3.20–8.11)	<0.001
JAK2	0.83	(0.65–1.06)	0.140
CALR	0.49	(0.29–0.83)	0.009
Splenomegaly ^+^	1.12	(0.80–1.58)	0.509
MPN-SAF (9–31.5) ***	0.98	(0.19–5.28)	0.985
MPN-SAF (≥31.5) ***	0.60	(0.05–5.82)	0.600
Ruxolitinib	0.04	(0.01–0.12)	<0.001

PPV-MF, myelofibrosis secondary to polycythemia vera. PET-MF, myelofibrosis essential thrombocythemia. Other abbreviations are explained in Table 1. * Reference category: <61.4 years. ** Reference category: low risk. *** Reference category: <9. ^+^ Splenomegaly determined by imaging methods such as ultrasonography or computed tomography.

**Table 3 cancers-14-02331-t003:** Multivariate Cox’s proportional hazards models.

Variable	Without Ruxolitinib			With Ruxolitinib		
	HR	95% CI	*p*-Value	HR	95% CI	*p*-Value
Age (61.4–76.3 y) *	2.57	(1.12–5.61)	0.026	2.91	(1.24–6.85)	0.014
Age (≥76.3 y) *	4.85	(1.75–13.41)	0.002	4.23	(1.53–11.73)	0.006
Hypertension	0.99	(0.52–1.92)	0.992	1.13	(0.59–2.16)	0.707
Diabetes	2.20	(0.98–4.94)	0.057	1.46	(0.61–3.49)	0.394
Dyslipidemia	1.40	(0.68–2.87)	0.365	1.07	(0.48–2.36)	0.872
Renal dysfunction	2.08	(1.023–4.21)	0.043	1.69	(0.81–3.50)	0.159
Pulmonary disease	2.69	(1.47–4.91)	0.001	2.40	(1.29–4.47)	0.006
Smoking	3.34	(1.85–6.04)	<0.001	3.82	(2.02–7.24)	<0.001
HCV	11.49	(2.74–48.25)	0.001	9.86	(2.34–41.64)	0.002
IPSS Intermediate-1 **	2.53	(0.73–8.91)	0.142	2.94	(0.82–10.52)	0.096
IPSS Intermediate-2 **	4.76	(1.40–16.22)	0.013	6.90	(1.95–24.37)	0.003
IPSS High Risk **	11.34	(3.24–39.7)	<0.001	15.97	(4.22–60.42)	<0.001
CALR	0.97	(0.28–3.42)	0.966	0.82	(0.23–2.89)	0.763
Ruxolitinib	-	-	-	0.12	(0.04–0.43)	0.001

HR: hazard ratio. Other abbreviations are explained in Table 1. * Reference category: <61.4 years. ** Reference category: low risk.

## Data Availability

The data that support the findings of this study are available from the corresponding author, M.G.-F., upon reasonable request.

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
