# Peer review of "Impact of Individual Comorbidities on Survival of Patients with Myelofibrosis"

_cancers, 2022, doi:10.3390/cancers14092331_

Round 1

Reviewer 1 Report

Comments to the authors

Minor

  • References in the text need to be checked, many references are not in parentheses!
  • Please state, if there were also pat. with prefibrotic myelofibrosis included and what’s the percentage.
  • A table, which compares the disease characteristics of pat. with and without comorbidities should be included (supplementary)

Major

  • Ad study design and Table S1/ definition of comorbidities: in general, the definitions are not really strict which makes the conclusions weak. Especially the definition of smoking (“having smoked over 100 cigarettes in their lifetime and having smoked within the last 180 days” does not define a real smoker; e.g having smoked for 20 years but stopped a year ago should be counted as a smoker; smoking of 120 cig = 6 packs in total lifetime will likely not have an impact on diseases ), other neoplasms (why is e.g. early stage cervical cancer not excluded?), pulmonary disease (“dyspnea on slight activity-“ this could also be due to bad physical condition and is not always a sign of pulmonary disease). This issue should be more extensively discussed and is a major limitation of the study.
  • It might be a difference whether a comorbidity like diabetes, hypertension and dyslipidemia is controlled or not. Please include comments on that.

Author Response

We are grateful for your time and constructive comments on our manuscript. Below, we also provide a point-by-point response explaining how we have addressed each of your comments.

Minor:

References in the text need to be checked, many references are not in parentheses.

- We have corrected the references style at the manuscript.

Please state, if there were also pat. with prefibrotic myelofibrosis included and what’s the percentage.

- We don’t diferenciate patients with prefibrotic myelofibrosis. The inclusión criteria of our study accepted any mylofibrosis diagnosis by WHO criteria.

The WHO 2016 included the prefibrotic myelofibrosis term to detect the wrong diagnosis of essential trombocythemia on patients who really had a prefibrotic myelofibrosis. So, we didn’t consider it could be relevant for the aims of our study.

A table, which compares the disease characteristics of patients with and without comorbidities should be included (supplementary).

-The aim of our study is the impact of the individual comorbidities rather than the comorbidity burden vs non-comorbidities. In addition, our patients had a median age of 68 years so there was only just a few patients without any comorbidity. On the other hand, the comorbidity burden have been previous studied by other groups, and for this reason we focused in the prevalence of individual comorbidities.

Major:

Ad study design and Table S1/ definition of comorbidities: in general, the definitions are not really strict which makes the conclusions weak. Especially the definition of smoking (“having smoked over 100 cigarettes in their lifetime and having smoked within the last 180 days” does not define a real smoker; e.g having smoked for 20 years but stopped a year ago should be counted as a smoker; smoking of 120 cig = 6 packs in total lifetime will likely not have an impact on diseases ), other neoplasms (why is e.g. early stage cervical cancer not excluded?), pulmonary disease (“dyspnea on slight activity-“ this could also be due to bad physical condition and is not always a sign of pulmonary disease). This issue should be more extensively discussed and is a major limitation of the study. It might be a difference whether a comorbidity like diabetes, hypertension and dyslipidemia is controlled or not. Please include comments on that.

- Smoking: Current smokers are defined in the National Health Interview Survey (NHIS), as a person who reports currently smoking tobacco. NHIS also requires a current smoker to have smoked at least 100 cigarettes (5 packs) in his or her lifetime.

Former smokers are defined in NHIS, a person who does not currently smoke tobacco but has smoked at least 100 cigarettes in his or her lifetime. Because relapse to smoking occurs frequently after quitting, long-term abstinence is often operationally defined as 6 months of abstinence. Abstinence from smoking for at least 7 days in a row is the criterion often required in clinical studies for an individual to be considered a former smoker in the short-term.

So we decided follow the NHIS Current Smoker definition although we also included the patients that could be former smokers by NHIS but they have smoked within the last 6 months (we decided it because relapse to smoking occurs frequently after quitting, long-term abstinence is often operationally defined as 6 months of abstinence).

In addition internationals validate models to evaluate the cardiovascular risk, as SCORE (Systematic Coronary Risk Evaluation) or Framingham Risk Score, include the smoking habit and they only consider if the patient is current smoker or not.

- Other neoplasms: we have followed the definition of validate comorbidity scores as Charlson  Comorbidity Index (CCI) or The Hematopoietic cell transplantation specific comorbidity index (HCT-CI), both widely used for hematologicals diseases.

- Pulmonary disease:  We have detailed the definition at the document (Supplementary S1).          

- About if might be a difference whether a comorbidity like diabetes, hypertension and dyslipidemia is controlled or not Please include comments on that. 

We totally agree. For this reason we included comments on that at the first versión of the manuscript. At the discussion you can find “Several recent studies on the impact of cardiovascular risk factors (CVRF) such as diabetes, hypertension or dyslipidemia on cardiovascular complications have demonstrated the significant improvement in outcomes achieved by early, effective control.[…] Therefore, close collaboration between the hematologist and other medical specialists in managing and controlling comorbid conditions is an important aspect of achieving optimal outcomes.” In addition, we specify in the limitations that “the impact of comorbidities may change due to variations in the medical care provided”.

Reviewer 2 Report

Comorbidity burden is an important risk factor for overall survival (OS) in several hematological malignancies. This prospective study was conducted to evaluate the impact of individual comorbidities on survival in 668 patients with primary myelofibrosis (PMF), secondary to polycythemia vera (PPV-MF) or essential thrombocythemia (PET- MF). The results showed that pulmonary disease, smoking, renal dysfunction and hepatitis C virus exhibited negative impacts on OS. Treatment with ruxolitinib reduced the deleterious effect of specific comorbidities. Therefore, the authors proposed that individual comorbidities should be taken into account in determining the survival prognosis for patients with myelofibrosis. With rigorous data processing, this article was reasonably designed. However, the clinical analysis lacked novelty and feature. Due to the competitive nature of the articles submitted to this journal, the paper does not meet the criteria for publication. Relevant published papers are listed as follows:

[1] Bartoszko J, Panzarella T, McNamara CJ, Lau A, Schimmer AD, Schuh AC, Sibai H, Maze D, Yee KWL, Devlin R, Gupta V. Distribution and Impact of Comorbidities on Survival and Leukemic Transformation in Myeloproliferative Neoplasm-Associated Myelofibrosis: A Retrospective Cohort Study. Clin Lymphoma Myeloma Leuk. 2017; 17: 774-781.

Author Response

We are grateful for your time and constructive comments on our manuscript. Below, we also provide a point-by-point response explaining how we have addressed each of your comments.

Review Report: “…Therefore, the authors proposed that individual comorbidities should be taken into account in determining the survival prognosis for patients with myelofibrosis. With rigorous data processing, this article was reasonably designed. However, the clinical analysis lacked novelty and feature. Due to the competitive nature of the articles submitted to this journal, the paper does not meet the criteria for publication. Relevant published papers are listed as follows:

[1] Bartoszko J, Panzarella T, McNamara CJ, Lau A, Schimmer AD, Schuh AC, Sibai H, Maze D, Yee KWL, Devlin R, Gupta V. Distribution and Impact of Comorbidities on Survival and Leukemic Transformation in Myeloproliferative Neoplasm-Associated Myelofibrosis: A Retrospective Cohort Study. Clin Lymphoma Myeloma Leuk. 2017; 17: 774-781.”

- Thank you very much for appreciate our work. However, I would like to explain that, as we detail in the introduction, our study is the first that analize the impact of individual comorbidities on MF patients.

“Although few studies have assessed the impact of comorbidities on the outcome of patients with MF, retrospective studies have shown that a high burden of comorbidities as defined by the Adult Comorbidity Evaluation-27 (ACE-27) [31] is significantly associated with an unfavorable impact on survival in MF [32–34]. However, these studies considered the cumulative comorbidity burden rather than the impact of individual comorbidities on survival. The main aim of the present study is to assess the influence of individual comorbidities on survival in a prospective multicenter series of MF patients.”

In the same way, during the discussion we even referred the article that you are citing, and we explain how this and others articles analized the comorbidity burden of MF patients but no one studied de impact of individual comorbidities on MF survival. “Comorbidities are known to be associated with inferior survival among patients with MF [32–34]. The ACE-27 [31], which is specifically designed for patients with cancer, is the instrument most commonly used in previous studies of MF patients to measure the severity of their comorbidities [32–34]. In 2014, a retrospective study of 131 MF patients by Lekovic et al. [34] suggested ACE-27 could help predict patient survival. On the other hand, their multivariate model did not find age to be an important prognostic factor for survival. In the same year, Newberry et al. [33] reported that comorbidities had a significant negative impact on OS in PMF patients aged < 65 years, but not in older patients. More recently, Bartozsko et al. [32] evaluated two comorbidity scales in a cohort of 309 patients with PPV-MF or PET-MF and PMF, and concluded that severe comorbidities according to ACE-27 (score ≥ 3) were associated with a reduced OS. By contrast, a high score on the Hematopoietic Cell Transplantation Comorbidity Index (HCT-CI) [36] did not reach statistical significance. No differential effect of severe comorbidities on survival based on age was detected. Finally, Breccia et al. [37] showed that baseline comorbidities did not influence the probability of achieving spleen/symptom responses for MF patients receiving ruxolitinib treatment. It should be noted that the assessment of comorbidities differed among these studies, which precludes our reaching firm conclusions on the impact of this factor on survival in MF.”

Reviewer 3 Report

This GEMFIN study aims to describe the effect of comorbidities on OS in MF (n = 668). Comorbidities have been shown to affect survival in related heme malignancies, like MDS and CML, but are notably excluded from current MF prognostic models. The authors on MVA find that lung disease, smoking, renal dysfunction and HCV adversely impact OS, as do advanced age and higher risk IPSS (as is well known), and a favorable impact of rux (this has been shown in many other studies). In fact, when rux is included as a variable, the significance of renal dysfunction disappears, which is interesting because rux has been reported to improve renal function in MF. The authors suggest that the OS benefit of rux is tied to its anti-inflammatory effects.

Question for the authors: regarding the UVA, both in the abstract and in the text, they separately list HTN, dyslipidemia, smoking and HCV as being adversely associated with OS, and then list DM, lung and renal disease as being "also associated with an increased risk of death". But the p values are the same or very similar, so the language is a bit confusing. Are the authors trying to draw some sort of distinction between the first 4 factors and the last 3 factors? 

Author Response

We are grateful for your time and constructive comments on our manuscript. Below, we also provide a point-by-point response explaining how we have addressed each of your comments.

- Review Report: 

Regarding the UVA, both in the abstract and in the text, they separately list HTN, dyslipidemia, smoking and HCV as being adversely associated with OS, and then list DM, lung and renal disease as being "also associated with an increased risk of death". But the p values are the same or very similar, so the language is a bit confusing. Are the authors trying to draw some sort of distinction between the first 4 factors and the last 3 factors?

- We explain how the first 4 factors (Hypertension, Dyslipidemia, Smoking and HCV) were the “most adversely associated with OS” because all of them had HR near to 5 with statiscal significance. Then, we said that the others 3 factors (DM, pulmonary and renal disease) were “also associated with an increased risk of death” because, even every one got statistical significance, the HR were about 2 or 3.

Round 2

Reviewer 1 Report

I appreciate the detailed answers to the concerns.

Please also include them into the manuscripts. Especially 2 things are important:

1. how many (number, percentage) of the pat. had prefibrotic PMF. This is important to know for the reader since this pat. have a different survival and it could be that Comorbidities do have a different impact in this patients. In case you do not have this number you should state that.

2. A more detailed information how comorbidities where defined should be given in the methods section.  

Author Response

First, thank you again for your comments.

1. how many (number, percentage) of the pat. had prefibrotic PMF. This is important to know for the reader since this pat. have a different survival and it could be that Comorbidities do have a different impact in this patients. In case you do not have this number you should state that.

We agree with the reviewer that knowing how many patients had prefibrotic MF could be interesting but we didn’t include this data due to the study was designed before the 2016 WHO classification was published. Thus, we have added this point as a limitation of the study.

2. A more detailed information how comorbidities where defined should be given in the methods section.

We specify in methods section “see the Supplementary Appendix for a list of comorbidity definitions”, anyway we could add the table of the Supplementary Material (Table S1) in the methods section.